# Changes in early abortion access among out-of-state abortion patients in Illinois, following public insurance coverage through state Medicaid: A brief research report

**Carmela Zuniga**[1]*, **Terri-Ann Thompson**[1], **Danielle Young**[2], **Hanz Dismer**[3], **Lee Hasselbacher**[4], **Debra Stulberg**[4]

1 Ibis Reproductive Health, Cambridge, Massachusetts, United States of America, 2 School of Social Work and Social Policy, Trinity College Dublin, College Green, Dublin, Ireland, 3 Hope Clinic, Granite City, Illinois, United States of America, 4 Department of Family Medicine, University of Chicago, Chicago, Illinois, United States of America

* czuniga@ibisreproductivehealth.org

## Abstract

In 2018, Illinois implemented House Bill 40 (HB40) which required Medicaid (means-tested public insurance) coverage of abortion care for Illinois residents. Medicaid coverage of abortion increases financial accessibility, which oftentimes leads to earlier access to care for covered patients. The ability of residents to use Medicaid may have increased the availability of financial assistance for non-residents. However, whether Medicaid coverage is associated with any changes in abortion access among out-of-state patients – who cannot use Medicaid for abortion coverage – is unknown. To explore if Medicaid coverage of abortion is associated with changes in abortion access for out-of-state patients, we analyzed de-identified records of abortion visits among non-Illinois residents presenting for abortion care across 12 Illinois health centers. We used logistic regression to assess if presenting early for an abortion (≤11 weeks gestation) was associated with implementation of HB40 (2017 vs 2018–2019). Although out-of-state residents were more likely to present early if they received abortion care in the post-HB40 period than pre-HB40 (81% at ≤11 weeks in 2018–2019 vs 78% in 2017), multivariable regression shows that HB40 was not associated with early abortion access when controlling for other patient characteristics. Out-of-state patients had higher odds of presenting ≤11 weeks of gestation during the study period if they were a resident of a state bordering Illinois (OR 1.89, 95% CI 1.55-2.30, p<0.001) or were over the age of 24 (OR 1.50, 95% CI 1.35-1.66; p<0.001). Given the significant rise in out-of-state patients after the *Dobbs* decision, future research should explore how large increases in patient volume over time have impacted abortion care in Illinois and other states, and assess how federal and state-level legal or policy changes influence abortion access for out-of-state patients.

**Data availability statement:** This work analyzed patient-level data from two large clinics, Hope Clinic and Planned Parenthood of Illinois, which provide abortion care in Illinois, United States of America, under approved Data Use Agreements. These agreements prohibit data sharing to protect subject privacy. Our research team did not receive special privileges from these collaborating clinics. The clinics review requests to partner on research and share data on a case-by-case basis. Researchers wishing to collaborate with these clinics for research can contact them at: 1) info@hopeclinic.com or 618-451-5722 2) research@ppil.org or 312-592-6884.

**Funding:** This work was supported by the Irving Harris Foundation (unnumbered grant to LH) and by a research grant from the Susan Thompson Buffett Foundation (unnumbered grant to TAT).

**Competing interests:** The authors have declared that no competing interests exist.

## Introduction

Medicaid is a public health insurance program for people earning low incomes in the United States. Though jointly funded by federal and state governments, Medicaid is mostly managed by states, resulting in state-level variation in eligibility requirements and services covered. Residents are generally unable to use Medicaid to cover the cost of health care services provided in other states. In 2018, the state of Illinois passed House Bill 40 (HB40), making Illinois one of two Midwestern states where abortion services are covered by Medicaid [1,2]. Medicaid coverage of abortion in this state is associated with decreased out-of-pocket costs [3,4] and an increase in abortion volume [5], suggesting increased abortion access for Medicaid recipients. However, out-of-state abortion patients who cannot use Medicaid have also been contributing to increased patient volume in Illinois. Between 2017 and 2019, the number of out-of-state abortion patients increased by 36%, compared to a 18% increase among residents during this same period, and has increased every year since [6]. Large increases in abortion patient volume may result in patients having abortions at later gestational ages due to potentially longer wait times between scheduling and having an appointment. One study found that gestational age at the time of abortion increased by a median of eight days after HB40 [3] and another study found that the time between scheduling and having an abortion increased by about 3 days [7]. The latter study also found an overall increase in gestational age after HB40, though there was a decrease in gestational age among Medicaid patients in particular [7].

There is limited evidence suggesting that Medicaid coverage may indirectly lead to earlier abortion access for patients without Medicaid in Illinois. One study exploring the perspectives of providers and community stakeholders on HB40 found that some participants thought the law allowed resources to be shifted from Medicaid patients to other low-income patients that do not have or cannot use their insurance [8]. This potential increase in the availability of funding for some out-of-state patients may reduce the time needed to gather funds for their abortion and travel expenses, and could contribute to earlier gestational ages at the time of abortion. In addition, patients presenting for their abortion at up to 11 weeks gestation can opt to have a medication abortion and have their abortion at home, whereas those presenting at later gestational ages can only have an in-office procedural abortion [9]. Given the limited evidence on associations between Medicaid coverage and abortion access for out-of-state patients, we aimed to examine changes in early presentation for abortion (≤11 weeks gestation) for out-of-state patients one year before and two years after the implementation of HB40.

## Materials and methods

### Ethics statement

This study was approved by the University of Chicago Institutional Review Board (IRB20–0901). We were granted a waiver of informed consent from the Institutional Review Board due to the following reasons: this was a minimal-risk study; the research could not have been feasibly carried out without the waiver because

we were not collecting names or contact information and this would have increased the risk of loss of confidentiality; the waiver would not have adversely affected the rights and welfare of the subjects involved in the research.

**Data.** We emailed all health centers in Illinois offering medication and/or procedural abortions and that provided more than 500 abortions annually to ask if they were interested in participating in the study. A large independent clinic and fifteen clinics affiliated with one organization met these criteria and were able to share data on abortion visits that occurred from 2017 to 2019. Twelve of the affiliated clinics provided abortion services in 2017, and by 2019, three existing clinics began providing medication abortion services and a new affiliated clinic opened that provided both medication and procedural abortion services. The large independent clinic is located in southern Illinois at the border of Missouri and the majority of their abortion patients at the time of the study were not Illinois residents.

Though not all abortions are reported to the Illinois Department of Public Health (IDPH), we used IDPH data to estimate the proportion of abortions captured in our sample because IDPH reports the total number of abortions provided for each year of our study, as well as the number of abortions provided to out-of-state patients. Compared to annual abortion data collected from the Guttmacher Institute (which gathers abortion data from a variety of sources, including state health departments) [10], IDPH annual data on the total number of abortions was 6.5% and 11% lower for 2017 and 2019, respectively [6]. Dividing the annual abortion totals of our sample by the annual abortion totals reported by the IDPH for each year of the study [6], we estimate these health centers covered between 53%-58% of all abortion care reported in the state. Based on the state-wide number of out-of-state visits reported by IDPH for each year of this study, our sample represents approximately 55% of all out-of-state abortion patients between 2017 and 2019.

Data were cleaned and merged into one dataset using R. If data on state of residency were missing, we used reported zip code data to determine patients' state of residency and included all patients living in states other than Illinois. Gestational age was determined by ultrasound at all health centers and gestational ages reported in days were converted to weeks. Less than 1% of our abortion visit data was missing information on gestational age and less than 4% was missing data on patient race. There were no missing data on age or state of residency.

**Outcomes and measures.** Our primary outcome was early access to abortion, defined as presenting at ≤11 weeks gestation, as this is the cut-off time for being able to choose to have a medication abortion. We also report on changes in out-of-state and resident patient volume between 2017 and 2019 at each clinic, as well as changes in the share of out-of-state and resident abortion patients seen across clinics.

**Statistical analysis.** We conducted a bivariate logistic regression analysis to assess the association between early access to abortion and time (pre-HB40 vs post-HB40). As some abortion providers were delayed in implementing HB40 when it went into effect in 2018 [11], we included 2019 data in our post-HB40 analysis. We also conducted multivariable regression analysis with a random intercept on clinics to account for clinic-level changes and adjusted for out-of-state patient characteristics (race, age, and state of residency) in our model. Race was included as a covariate because at the time of the study, the majority of abortion patients nationwide were women of color [12,13] and recent data suggest that compared to white women, women of color are more likely to have low incomes and lack access to a vehicle [14]. With limited resources and transportation options, it may take women of color longer to gather sufficient funds for abortion care and finalize travel logistics, which may result in later gestational ages at the time of abortion. Similarly, young people may also find it difficult to gather sufficient funds for abortion care and travel, so we created a binary variable for age (≤24 years and >24 years) and added this as a covariate. We also created a binary variable for state proximity to Illinois (states bordering Illinois versus not bordering Illinois), as the logistical and financial barriers of traveling out-of-state may be less burdensome for residents of bordering states than residents in states not bordering Illinois. We used R to complete all analyses.

Four affiliated clinics, which provided 5% of total visits and 2% of out-of-state visits in our dataset, did not provide abortions during the pre-HB40 period so we could not calculate changes in volume for these clinics and these clinics were not included in analyses comparing differences between the pre- and post-HB40 periods or in our multivariable regression

analysis. We checked whether including data from these four clinics impacted our analysis and found that their inclusion did not change our overall regression results. Abortion visits at these four clinics are included when we report on patient characteristics and overall changes in patient volume and distribution over the study period.

## Results

From a total of 59,550 abortion visits across 16 health centers between 2017–2019, our out-of-state dataset included 10,298 abortion visits at 15 health centers by 9,377 unique patients. Most out-of-state abortion visits were made by patients from the border states of Missouri (69.8%), Indiana (16.1%), and Wisconsin (5.1%).

Table 1 displays out-of-state patient characteristics from 2017 to 2019. Between 2017 and 2019, there was an increase in the proportion of out-of-state patients coming from Missouri (61.9% to 75.5%) and small increases in patients identifying as Black or African American and being over the age of 24.

### Changes in overall patient volume and distribution

Between 2017 and 2019, total volume of abortion patients (both residents and out-of-state patients) across all 16 clinics in our sample increased by 28% (from 17,373 in 2017–22,318 in 2019). Across the 12 clinics providing abortions prior to HB40, there was a 15% increase in abortion patients. Looking at volume increase by state of residency across these 12 clinics (n = 10,049) during the same period, the volume of abortion visits made by Illinois residents increased by 11% while

**Table 1. Out of-state patient demographics, 2017-2019 [a.]**

|  | Overall | 2017 | 2018 | 2019 |
|---|---|---|---|---|
|  | 10298 | 3052 | 3032 | 4214 |
| **Age Groups** | n (%) |  |  |  |
| ≤24 years | 3524 (34.2) | 1100 (36.0) | 1062 (35.0) | 1362 (32.3) |
| 13-17 | 287 (2.8) | 102 (3.3) | 99 (3.3) | 86 (2.0) |
| 18-24 | 3237 (31.4) | 998 (32.7) | 963 (31.8) | 1276 (30.3) |
| >24 years | 6774 (65.8) | 1952 (64.0) | 1970 (65.0) | 2852 (67.7) |
| 25-34 | 5131 (49.8) | 1493 (48.9) | 1492 (49.2) | 2146 (50.9) |
| 35-44 | 1618 (15.7) | 453 (14.8) | 474 (15.6) | 691 (16.4) |
| >44 | 25 (0.2) | 6 (0.2) | 4 (0.1) | 15 (0.4) |
| **Race** |  |  |  |  |
| White | 4439 (44.8) | 1326 (45.3) | 1363 (46.1) | 1750 (43.5) |
| Black or African American | 4779 (48.3) | 1389 (47.4) | 1390 (47.0) | 2001 (49.8) |
| Asian, Native Hawaiian, and/or Pacific Islander | 280 (2.8) | 84 (2.9) | 80 (2.7) | 116 (2.9) |
| American Indian or Alaska Native | 20 (0.2) | 7 (0.2) | 3 (0.1) | 10 (0.2) |
| Other race | 384 (3.9) | 122 (4.2) | 119 (4.0) | 143 (3.6) |
| **State of residence** |  |  |  |  |
| Missouri | 7190 (69.8) | 1888 (61.9) | 2119 (69.9) | 3183 (75.5) |
| Indiana | 1654 (16.1) | 645 (21.1) | 461 (15.2) | 548 (13.0) |
| Wisconsin | 524 (5.1) | 219 (7.2) | 168 (5.5) | 137 (3.3) |
| Kentucky | 189 (1.8) | 57 (1.9) | 63 (2.1) | 69 (1.6) |
| Iowa | 121 (1.2) | 36 (1.2) | 44 (1.5) | 41 (1.0) |
| Michigan | 79 (0.8) | 38 (1.2) | 19 (0.6) | 22 (0.5) |
| Non-bordering states | 541 (5.3) | 169 (5.5) | 158 (5.2) | 214 (5.1) |

a This table reflects abortion visits and not individual patients, so patients who received more than one abortion are represented more than once.

*There is no race data reported for 3.8% of visits between 2017–2019.

the volume of visits made by out-of-state patients increased by 33%. Changes in patient volume varied at each of the 12 health centers between 2017 and 2019, with five experiencing volume increase of over 10%, two experiencing a decrease of over 10%, and five seeing little or no change in volume (0–10% change) (Table 2).

In addition to looking at changes in patient volume, we also looked at shifts in the share of out-of-state and resident abortions provided by clinics to identify any changes in the distribution of patients across clinics (S1 Table). We found little variation in the distribution of out-of-state and resident abortions across the twelve clinics between 2017 and 2019, with most clinics providing care to about the same percentage of out-of-state and resident abortions (+/- 2 percentage points). The two exceptions were the independent clinic, which experienced an 18-percentage point increase in out-of-state abortions and a 5-percentage point increase in resident abortions, and Clinic 8, which experienced a 12-percentage point decrease in out-of-state abortions and a 6-percentage point decrease in resident abortions.

## Presenting for an abortion at ≤11 weeks gestation

Across the 12 clinics providing abortions prior to HB40, the number of out-of-state patients receiving an early abortion increased by 40%. Comparing pre- vs post-HB40 periods, the proportion of out-of-state patients receiving care at ≤11 weeks gestation across 12 clinics increased from 77% in 2017 to 80% after HB40 was implemented (2018–2019). Bivariate logistic regression analysis reveals increased odds of out-of-state patients being at ≤11 weeks gestation in the post-HB40 period than in 2017 (OR 1.16, 95% CI 1.05 -1.30, p = 0.004). However, time was no longer significantly associated with presenting early for an abortion when taking into account clinic-level variations, patient race (white vs people of color), patient age (<25 vs >=25 years), and patient state of residency (bordering vs not bordering Illinois). Out-of-state patients across 12 clinics had higher odds of presenting ≤11 weeks of gestation during the study period if they were a resident of a state bordering Illinois (OR 1.89, 95% CI 1.55-2.30, p < 0.001) or were over 24 years old (OR 1.50, 95% CI 1.35-1.66; p < 0.001). Table 3.

**Table 2.** Changes in volume out-of-state and IL resident abortion patients at each clinic between 2017-2019*.

| Clinic | OOS | | | Residents | | | All patients | | |
|---|---|---|---|---|---|---|---|---|---|
| | 2017 n | 2019 n | % Change | 2017 n | 2019 n | % Change | 2017 n | 2019 n | % Change |
| Clinic 1 | 136 | 132 | -2.9 | 3098 | 3244 | 4.7 | 3234 | 3376 | 4.4 |
| Clinic 2 | 3 | 8 | 166.7 | 329 | 333 | 1.2 | 332 | 341 | 2.7 |
| Clinic 3** | 4 | 24 | 500.0 | 51 | 430 | 743.1 | 55 | 454 | 725.5 |
| Clinic 4** | 34 | 16 | -52.9 | 434 | 513 | 18.2 | 468 | 529 | 13.0 |
| Clinic 5 | 13 | 5 | -61.5 | 205 | 113 | -44.9 | 218 | 118 | -45.9 |
| Clinic 6 | 12 | 5 | -58.3 | 274 | 220 | -19.7 | 286 | 225 | -21.3 |
| Independent clinic | 1964 | 3354 | 70.8 | 1059 | 1926 | 81.9 | 3023 | 5280 | 74.7 |
| Clinic 7 | 112 | 79 | -29.5 | 1376 | 1408 | 2.3 | 1488 | 1487 | -0.1 |
| Clinic 8 | 577 | 294 | -49.0 | 5653 | 5379 | -4.8 | 6230 | 5673 | -8.9 |
| Clinic 9 | 22 | 22 | 0.0 | 355 | 460 | 29.6 | 377 | 482 | 27.9 |
| Clinic 10 | 111 | 63 | -43.2 | 799 | 848 | 6.1 | 910 | 911 | 0.1 |
| Clinic 11 | 64 | 70 | 9.4 | 688 | 1078 | 56.7 | 752 | 1148 | 52.7 |
| **All clinics** | **3052** | **4072** | **33.4** | **14321** | **15952** | **11.3** | **17373** | **20024** | **15.3** |

*Experienced a total patient volume increase >10%.

** Excludes 3 affiliated clinics that did not provide abortions in 2017 and 1 affiliated clinic that did not open until 2018.

**Table 3. Variables associated and not associated with receiving care at ≤11 weeks gestation among out-of-state patients.**

| | Odds ratio (95% CI) | p-value |
|---|---|---|
| **Bivariate model** | | |
| Post HB-40 (2018–2019) | 1.16 (1.05-1.29) | 0.004 |
| **Multivariable model** | | |
| Post HB-40 (2018–2019) | 1.07 (0.96-1.19) | 0.23 |
| Resident of bordering state | 1.89 (1.55-2.30) | <0.001 |
| Older than 24 years | 1.50 (1.35-1.66) | <0.001 |
| Racial identity: white | 1.09 (0.98-1.20) | 0.12 |

## Discussion

Although our study did not find an association between Medicaid coverage of abortion and early abortion access for out-of-state patients, we found that the odds of presenting early for an abortion increased for residents of bordering states and patients over 24 years old. It is possible that residents of states bordering Illinois have less distance to travel compared to residents of non-bordering states, so travel-related costs (accommodation, gas, time taken off work, childcare for existing children) may be less financially and logistically burdensome for patients geographically closer to Illinois. However, overcoming financial and logistical barriers may be especially difficult for younger patients and our results show that out-of-state patients 24 years old or younger were less likely than their older counterparts to present at ≤11 weeks of gestation during the study period. Research has shown that young patients encounter unique information and logistical barriers to accessing abortion care [15], and that many patients traveling out of state may not want to disclose their location or abortion intentions to family and friends [16]. Though disclosing one's abortion plan to friends and/or family could result in financial and logistical support [17], young people unwilling to do so may have to delay care to gather funds for lodging and transportation costs in addition to funds for abortion care.

Another possible reason for earlier abortion access among out-of-state patients may be due to an increased availability of funding for these patients. However, more research is needed to understand if there was a significant reallocation of funds to out-of-state patients and if such funding contributed to earlier abortion access.

The increased number of out-of-state patients receiving an abortion at early gestational ages despite increases in total patient volume may have been due to increased clinic staffing, expanded provision of abortion services by clinics in our sample, and/or the opening of additional health centers offering abortion services [18]. Additional facilities and staffing likely made it easier to meet the increased demand for abortion care and may have allowed a higher proportion of patients to be seen at earlier gestational ages.

Similar to the data recorded by IDPH, our data also revealed a larger percentage increase in out-of-state abortion visits (33%) than visits by Illinois residents (11%)between 2017 and 2019. During this time period there was an increase in Missourians receiving care in Illinois, which may have been driven by two Missouri-specific events that occurred during the study period. The first occurred in October 2018, when new state requirements for abortion providers led to Missouri having only one abortion clinic [19]. The following year, Missouri passed a law banning abortions after 8 weeks [20]. It is possible that some of the overall increase in volume of out-of-state patients in our dataset can be attributed to Missouri state policies, making it difficult to assess the indirect impact of HB40 on Missourians and other out-of-state patients.

Although our sample captures almost 60% of out-of-state abortions provided in the state, it does not capture all abortion providers so our results may not be applicable to all out-of-state patients. In addition, not all abortions are reported to the IDPH, so our estimate of the proportion of abortions in our sample may be overestimated.

It is important to note that our data was collected before the 2022 *Dobbs v Jackson Women's Health Organization* decision, which allowed states to ban or severely restrict abortion access [21]. Since the decision, out-of-state travel has varied nationwide, with increases in the first years following the decision and slight decreases in the latter following a surge in telehealth provision [22]. However, overall out-of-state travel is substantially higher across the country since the decision [23], and Illinois has continued to experience a large increase in of out-of-state abortion patients, with 39% of all abortions provided to out-of-state patients in 2024 [22]. This may be due to the fact that Illinois borders three states which have banned abortion [24], is one of the few states in the Midwest region of the U.S. where abortion is recognized as a right under the state constitution [25], and allows for later abortions (abortions obtained at or after 21 weeks gestation) [25,26]. Future research could explore how clinics have handled the increase in out-of-state abortion patients and the ways in which abortion funding in Illinois has responded to significant legal or policy changes, including HB40, the *Dobbs* decision, and the cuts to Medicaid funding outlined in the *2025 Reconciliation Bill*.

## Conclusions

Although we did not find an association between Medicaid coverage of abortion for Illinois residents and earlier abortion access for out-of-state patients, our findings highlight that out-of-state patients are more likely to receive early abortion care if they reside in a border state and if they are older than 24 years. Future research should assess the factors that facilitate abortion access for out-of-state patients, especially adolescents and young adults. Such research is especially critical after the 2022 *Dobbs v Jackson Women's Health Organization* decision, which has resulted in an increasing number of out-of-state abortion patients. Though our study suggests increased volume of patients did not translate to later abortion access for out-of-state patients, it remains unclear how very large increases in out-of-state patients over time impacted abortion care in Illinois and other states, and how clinics and abortion funds have handled the increase in demand. Finally, more studies are needed that assess direct and indirect impacts of supportive abortion policies, such as Medicaid coverage of abortion, on abortion access for those residing in and out of state.

## Supporting information

**S1 Table.  Change in distribution of out-of-state and in-state abortion visits across clinics between 2017 and 2019.** (XLSX)

## Acknowledgments

The authors acknowledge the work of clinic, health center, and research staff who supported data extraction.

## Author contributions

**Conceptualization:** Carmela Zuniga, Terri-Ann Thompson, Lee Hasselbacher, Debra Stulberg.

**Data curation:** Carmela Zuniga, Danielle Young, Hanz Dismer, Lee Hasselbacher.

**Formal analysis:** Carmela Zuniga.

**Funding acquisition:** Terri-Ann Thompson.

**Methodology:** Terri-Ann Thompson, Danielle Young, Debra Stulberg.

**Resources:** Hanz Dismer.

**Supervision:** Terri-Ann Thompson, Debra Stulberg.

**Writing – original draft:** Carmela Zuniga.

**Writing – review & editing:** Terri-Ann Thompson, Danielle Young, Hanz Dismer, Lee Hasselbacher, Debra Stulberg.

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
