## [Decision Letter · Decision Letter 0]

11 Feb 2025

PGPH-D-24-02914

Changes in characteristics and gestational age of out-of-state abortion patients in Illinois, following public insurance coverage through state Medicaid: A brief research report

Dear Dr. Stulberg,

Thank you for submitting your manuscript to PLOS Global Public Health. After careful consideration, we feel that it has merit but does not fully meet PLOS Global Public Health’s publication criteria as it currently stands. Therefore, we invite you to submit a revised version of the manuscript that addresses the points raised during the review process.

Please note that we have only been able to secure a single reviewer to assess your manuscript. We are issuing a decision on your manuscript at this point to prevent further delays in the evaluation of your manuscript. Please be aware that the editor who handles your revised manuscript might find it necessary to invite additional reviewers to assess this work once the revised manuscript is submitted. However, we will aim to proceed on the basis of this single review if possible. 

We look forward to receiving your revised manuscript.

Kind regards,

Jianhong Zhou

Staff Editor

Journal Requirements:

1. Please provide an Author Summary. This should appear in your manuscript between the Abstract (if applicable) and the Introduction, and should be 150–200 words long. The aim should be to make your findings accessible to a wide audience that includes both scientists and non-scientists. Sample summaries can be found on our website under Submission Guidelines:

https://journals.plos.org/globalpublichealth/s/submission-guidelines#loc-parts-of-a-submission.

Additional Editor Comments (if provided):

Reviewers' comments:

Reviewer's Responses to Questions

**Comments to the Author**

1. Does this manuscript meet PLOS Global Public Health’s publication criteria?

Reviewer #1: No

2. Has the statistical analysis been performed appropriately and rigorously?

Reviewer #1: I don't know

3. Have the authors made all data underlying the findings in their manuscript fully available (please refer to the Data Availability Statement at the start of the manuscript PDF file)?

Reviewer #1: No

4. Is the manuscript presented in an intelligible fashion and written in standard English?

Reviewer #1: Yes

Reviewer #1: The data used in this analysis have the potential to provide useful insights into changes in characteristics of people traveling to IL for abortion care, but I don’t think they are appropriate for the research question undertaken by the authors. To understand how changes in Medicaid funding might have spillover effects, you would need some kind of data that speak to (changes in) how patients from out of state paid for this care—ideally from the patient records themselves. Data from changes in funding streams/allocations from abortion funds could also be useful.

Similarly, national trends show increases in medication abortion and abortion at earlier gestations. It’s not appropriate to attribute (even if tentatively) this to changes in funding. I could even pose the counterargument that, given that IL has always been a “haven” for abortions in the second and later trimester, the decline in these procedures among residents from other states is a potentially negative development. In reality, I think other dynamics are at work and the authors are over-interpreting their findings.

Abortion funding is not well understood even by many people in the US and needs to be better explained. For this study it would be useful to know (1) if all the clinics in the study worked with or had abortion funds (2) the scale of this funding and (3) if/how these dynamics varied across the clinics.

Finally, access to abortion care, and changes in it, in the neighboring states are not adequately addressed—and I’m not sure they could be. Did the increase in patients from MO crowd out availability for people from IN? Did the “missing” travelers go to another IL clinic that is not in the study? Did residents in IN and WI have better access to care in 2019 than 2017 and, in turn, have less need to travel to IL?

More generally, I’m not sure the article is written for a global audience, it assumes a lot of “insider” knowledge of Medicaid, US abortion laws and geography. (Notably, these shortcomings can be remedied easily.)

More specific feedback includes:

The abstract devotes too much text citing tenuous arguments (see next sentence) and needs to focus more on the data and findings. In contrast to Line, 6 the research I’m familiar with suggests that Medicaid/insurance coverage increases access to second trimester care, not that it leads to abortions at earlier gestations.

Intro: The authors need to be explicit when the research they are citing is specific to IL and not “global” or necessarily relevant to all states where Medicaid covers abortion care.

Methods. It is worth nothing that the ILHD undercounts abortions, and/or that not all abortions are reported to them compared to counts obtained by the Guttmacher Institute. This, in turn, means a potentially smaller share of abortions in the state are captured by the clinics in the study (assuming they all report to the HD).

I had to read the sentence spanning lines 62-66 several times to understand the differences in PI laws between the states. Consider revising to that this distinction/relative burden is clearer.

The sample is too large to take chisq seriously—everything tends to be statistically significant. Consider using some form of simple logistic regression instead.

Results

The stats in the first paragraph need context. I would expect abortion incidence to increase for IL residents in response to the change in Medicaid coverage so it’s hard to interpret the change in % of patients. Table 1 suggests a 38% increase but readers shouldn’t have to do the calculations by hand.

I initially looked for information on gestation and abortion type in Table 1, it’s not clear that this information is in the Figure. I’m not sure why you chose to display this information graphically and not in the table. The information in graphic format is a bit redundant since the measures are dichotomous.

Discussion

You should lead with something else, for example that after HB40 was implemented there was an increase in N of out of state patients among the clinics that participated in the study.

A number of abortion providers, and the majority of abortions provided, in the state are not included in the analysis. This needs to be addressed in the discussion. The patterns you uncovered may not be applicable to all people traveling from out of state.

The text in lines 155-157 goes in the Conclusions.

Conclusions: I generally consider the concluding section/paragraph a place for “loftier” policy implications. This one focuses on speculations and implications for future research. I recommend revisiting the content and organization of the last two sections (Discussion and Conclusions).

**Do you want your identity to be public for this peer review?** For information about this choice, including consent withdrawal, please see our Privacy Policy

Reviewer #1: **Yes: ** Rachel Jones

---

## [Decision Letter · Decision Letter 1]

19 Jun 2025

PGPH-D-24-02914R1

Changes in early abortion access among out-of-state abortion patients in Illinois following public insurance coverage through state Medicaid: A brief research report

Dear Dr. Stulberg,

Thank you for submitting your manuscript to PLOS Global Public Health. After careful consideration, we feel that it has merit but does not fully meet PLOS Global Public Health’s publication criteria as it currently stands. Therefore, we invite you to submit a revised version of the manuscript that addresses the points raised during the review process.

We look forward to receiving your revised manuscript.

Kind regards,

Farzana Kapadia

Academic Editor

Additional Editor Comments:

Dear Authors,

The reviewers and I agree that your paper is much improved. However, there are additional considerations and comments raised that should be addressed in this submission.

Thank you!

Reviewers' comments:

Reviewer's Responses to Questions

**Comments to the Author**

Reviewer #1: (No Response)

publication criteria?

Reviewer #1: Partly

3. Has the statistical analysis been performed appropriately and rigorously?

Reviewer #1: No

4. Have the authors made all data underlying the findings in their manuscript fully available (please refer to the Data Availability Statement at the start of the manuscript PDF file)?

Reviewer #1: No

5. Is the manuscript presented in an intelligible fashion and written in standard English?

Reviewer #1: Yes

Reviewer #1: This manuscript is substantially improved though the extensive revisions have introduced some new issues. In turn, I have some suggestions to make it a better paper.

Relatively minor but: You might consider de-identifying the clinics. Hope could be referred to as a large, independent clinic and you could list out the PP’s as Clinic 1 (or A), etc. There’s no need to reveal so much identifying information to external audiences (e.g., the abortion caseload at each individual clinic).

Abstract: “Impact” is too strong of a word for this analysis. You are looking to see if patterns in out-of-state folks suggest an impact. You are unable to assert causality.

Intro: the first paragraph doesn’t appropriately set up the study. The text focuses on Dobbs and the impact it has had on abortion incidence in IL, but the study is pre-Dobbs. Most of the information in the first paragraph should be moved to the Discussion.

The sentence on line 47-50 is a bit misfocused given that this analysis only goes through 2019—especially since in line 51 you revert back to the post-HB40 period. It might be more appropriate to indicate the proportion coming from oos during the study period, then note that this has increased substantially since Dobbs, making it even more important to understand these dynamics. (But also meaning they have limited usefulness given the recent surge, something you could address in the Discussion.) Also, this paragraph is very long and you might consider additional ways to break it up.

Methods: Line 73 needs clarification, presumably you emailed them asking them to participate in the study?

Lines 77-78: Be clear that these newly providing and new facilities are not included in this study. You can then remove the sentence on lines 121-123.

Line 82: you should provide one or more citations to back up this statement. Perhaps comparing CDC and Guttmacher figures for the same year? Readers might want a senses of how in/accurate the HD data are.

Section that starts in line 99: You need to explain which statistical methods you will be using.

Results: It’s confusing that the table uses different age categories than the text; for the latter, please provide the summary of over age 24. Or have a grouping in the table that has both larger and more detailed age groups.

Findings: Lines 157-161: It’s unclear why the figures in the text for Hope and Near North differ from those in the table. If this is not a mistake, please explain.

Line 163: It would be useful to have this information (<=11 weeks) in a Table.

Discussion: It’s inappropriate to lead with the slight increase in <=11 weeks among out of state patients given that you just told us it was qualified—that it disappears when you take other factors into account. In turn, you need to revise the rest of the sentence in the paragraph. The sentence/information starting on line 184 is hard to follow and this would also be better in the “shortcomings” section. Or in the paragraph that starts on line 198.

Line 195: provide a citation.

Paragraph starting on line 207: You might also consider citing Chiu et al.: https://www.mdpi.com/1660-4601/21/4/477

Conclusions: this section doesn’t really align with the study findings, it’s largely focused on how much we don’t know about the current situation. You should use this section to better contextualize the findings in the current context.

**Do you want your identity to be public for this peer review?** For information about this choice, including consent withdrawal, please see our Privacy Policy

Reviewer #1: **Yes: ** Rachel K Jones

---

## [Decision Letter · Decision Letter 2]

12 Oct 2025

PGPH-D-24-02914R2

Changes in early abortion access among out-of-state abortion patients in Illinois, following public insurance coverage through state Medicaid: A brief research report

Dear Dr. Stulberg,

Thank you for submitting your manuscript to PLOS Global Public Health. After careful consideration, we feel that it has merit but does not fully meet PLOS Global Public Health’s publication criteria as it currently stands. Therefore, we invite you to submit a revised version of the manuscript that addresses the points raised during the review process.

Please ensure you address all remaining concerns raised by the reviewer. 

We look forward to receiving your revised manuscript.

Kind regards,

Jen Edwards

Staff Editor

Journal Requirements:

Additional Editor Comments (if provided):

Reviewers' comments:

Reviewer's Responses to Questions

**Comments to the Author**

Reviewer #1: All comments have been addressed

publication criteria?

Reviewer #1: Yes

3. Has the statistical analysis been performed appropriately and rigorously?

Reviewer #1: Yes

4. Have the authors made all data underlying the findings in their manuscript fully available (please refer to the Data Availability Statement at the start of the manuscript PDF file)?

Reviewer #1: No

5. Is the manuscript presented in an intelligible fashion and written in standard English?

Reviewer #1: Yes

Reviewer #1: I appreciate the authors’ incorporation of my feedback. This paper is much improved, and I only have a few suggestions to make it better—and one correction that needs to be made.

Abstract: You might be explicit about how/why Medicaid coverage for IL residents would make abortion more accessible for out-of-state patients. You could potentially remove the 2nd or 3rd sentence and explain that the ability of residents to use Medicaid might have freed up financial assistance for non-residents.

Line. 69: The comparison of the IL HD and Guttmacher data is incorrect. While figures are not provided in the text, I see that Guttmacher reported 42080 and 52220 abortions in 2017 and 2019: https://onlinelibrary.wiley.com/doi/epdf/10.1363/psrh.12215.

By contrast, 39329 and 46517 abortions were reported to/by the IL HD (at least based on their published statistics): https://dph.illinois.gov/data-statistics/vital-statistics/abortion-statistics

These differences demonstrate that the HD stats are less complete and that this incompleteness may have increased/varied over time. Please revise the text accordingly. I suspect citation 11 is abortion by state of residence and the above PSRH citation is the more appropriate one.

Line 154: It might be useful to indicate *where Clinic 8 was located in the state in case it can provide insights into the increase (e.g., near a state border?).

Lines 159-163: not sure if you have the space, but it might be useful to have a table with this information. I had to read it a few times to make sure I understood what it was conveying.

Line 177: this is a run-on sentence.

Lines 185-191: These sentences could be condensed combined to make the text more efficient. I don’t know that you need to convey details like sample size in this context—you aren’t validating their findings, you’re just saying they suggest potential reasons for the associations you found. Ditto for lines 202-204, you don’t necessarily need to re-provide this detailed information; the point is that new clinics opened, including one that is not represented in the study.

Line 207-208: please revise so that it reads more clearly. Is 33% vs 11% the share of out of state patients in the two time periods in your data or the difference between your and IDPH data (or something else)?

Line 214: revise to read: indirect impact.

Lines 221-223: considering revising to make it clear that there is still substantially more travel (across the US) for abortion compared to pre-Dobbs.

**Do you want your identity to be public for this peer review?** For information about this choice, including consent withdrawal, please see our Privacy Policy

Reviewer #1: **Yes: ** Rachel Jones

---

## [Editor Report · Decision Letter 3]

22 Oct 2025

Changes in early abortion access among out-of-state abortion patients in Illinois, following public insurance coverage through state Medicaid: A brief research report

PGPH-D-24-02914R3

Dear Dr. Stulberg,

We are pleased to inform you that your manuscript 'Changes in early abortion access among out-of-state abortion patients in Illinois, following public insurance coverage through state Medicaid: A brief research report' has been provisionally accepted for publication in PLOS Global Public Health.

Best regards,

Julia Robinson

Executive Editor